# Ground-dwelling arthropods of pinyon-juniper woodlands: Arthropod community patterns are driven by climate and overall plant productivity, not host tree species

**Derek Andrew Uhey**[1]*, **Hannah Lee Riskas**[1], **Aaron Dennis Smith**[2], **Richard William Hofstetter**[1]

**1** School of Forestry, Northern Arizona University, Flagstaff, AZ, United States of America, **2** Department of Entomology, Purdue University, West Lafayette, IN, United States of America

* dau9@nau.edu

**Data Availability Statement:** All relevant data are within the manuscript and its Supporting Information files.

## Abstract

Pinyon-juniper (PJ) woodlands have drastically changed over the last century with juniper encroaching into adjacent habitats and pinyon experiencing large-scale mortality events from drought. Changes in climate and forest composition may pose challenges for animal communities found in PJ woodlands, especially if animals specialize on tree species sensitive to drought. Here we test habitat specialization of ground-dwelling arthropod (GDA) communities underneath pinyon and juniper trees. We also investigate the role of climate and productivity gradients in structuring GDAs within PJ woodlands using two elevational gradients. We sampled 12,365 individuals comprising 115 taxa over two years. We found no evidence that GDAs differ under pinyon or juniper trees, save for a single species of beetle which preferred junipers. Climate and productivity, however, were strongly associated with GDA communities and appeared to drive differences between sites. Precipitation was strongly associated with arthropod richness, while differences in GDA composition were associated with environmental variables (precipitation, temperature, vapor pressure, and normalized difference vegetation index). These relationships varied among different arthropod taxa (e.g. ants and beetles) and community metrics (e.g. richness, abundance, and composition), with individual taxa also responding differently. Overall, our results suggest that GDAs are not dependent on tree type, but are strongly linked to primary productivity and climate, especially precipitation in PJ woodlands. This implies GDAs in PJ woodlands are more susceptible to changes in climate, especially at lower elevations where it is hot and dry, than changes in dominant vegetation. We discuss management implications and compare our findings to GDA relationships with vegetation in other systems.

**Funding:** The author(s) received no specific funding for this work.

**Competing interests:** The authors have declared that no competing interests exist.

## Introduction

Pinyon pine (*Pinus edulis*) and juniper (*Juniperus* spp.) co-dominate woodlands that cover 19 million hectares in the southwestern United States [1]. Composition and distribution of pinyon-juniper (PJ) woodlands have drastically changed over the last century. Juniper has increased its range 10-fold, 'encroaching' into grassland/sagebrush habitats [2] and pinyon has experienced massive mortality from drought and subsequent bark beetle outbreaks [3, 4]. Droughts and temperature are expected to increase across the range of PJ woodlands, further altering this habitat [5, 6] and posing challenges for animals and conservation.

Animal species often specialize on host resources such as a specific host tree species or genus (i.e. the tree specialization hypothesis, a specific version of niche theory, [7]). The two dominant tree species in PJ woodlands offer distinct resources and habitat structures for animals. Pinyon and juniper differ drastically in composition of defensive secondary compounds [8] and wood properties [9, 10], resulting in distinct litter and soil under pinyon and juniper canopies [11, 12]. Additionally, junipers tend to provide more shelter through their shrub-like growth than pinyons resulting in higher moisture and cooler soil temperatures [13]. Despite these differences, comparisons between animal communities in pinyon and juniper are few. Bird assemblages differ between pinyon and juniper [14, 15] and other animal taxa may also have similar tree species preferences. If so, the shifting composition of PJ woodlands to primarily juniper may change animal communities and biodiversity within this ecotype.

Arthropods often closely associate with vegetation types [16] and can vary between tree species and even tree genotypes [17]. Most tests of tree species-effects on arthropods come from foliar-herbivore communities which live and feed directly on the living tree (e.g. [18, 19]). But unlike foliar communities, the majority of ground-dwelling arthropod (GDA) communities do not interact directly with living trees, instead harvesting nutrients from tree litter and microbes decomposing the litter, or preying on other arthropods, while using tree architecture for habitat [20]. GDA diversity is linked to litter composition mainly through diet preferences [21, 22] and canopy architecture via habitat preferences [23, 24].

The large diversity and cryptic habitats of GDAs have generally limited our understanding of their communities [25], and only a handful of studies have described GDA communities in PJ woodlands [26–28]. This lack of information is disconcerting because GDAs mediate above- and below- ground processes such as decomposition, soil aeration, and movement of soil nutrients [29] affecting tree regeneration and carbon storage [30]. Through their contribution to nutrient cycling and plant diversity, GDAs are critical to forest health [31]. Understanding and promoting GDAs in PJ woodlands may improve preservation and management of these ecosystems.

Climate can also be a strong driver of GDA assemblages [32], as temperature and precipitation regulate and limit arthropod physiology [33]. Arthropods are thermophilic, typically increasing in richness and abundance in warmer climates [34]. However, in the arid climates of PJ woodlands, the risk of desiccation from low precipitation and high temperatures (i.e. low vapor pressure) is often the limiting factor for arthropods [34]. Temperature and precipitation vary markedly across PJ ecosystems which are distributed across a large climate zone, often spanning over 1000m in elevation locally. Understanding how GDAs are distributed across these climatic gradients can inform what future GDA communities will look like after continued warming and drying.

Our study compares GDA communities in paired pinyon and juniper trees along two elevational gradients in northern Arizona over multiple seasons, encompassing large climate and productivity differences within the same biogeographical area. We test whether GDA communities differ between pinyon and juniper, and investigate the role of climate, productivity, and

season in structuring the aggregate PJ GDA communities. We analyze species-specific and whole GDA community patterns with corroborative statistical methods, giving a detailed examination of GDA responses to habitat and climate. In the face of biodiversity loss from climate change, documenting these patterns establishes a baseline for future comparisons and gives insight into the functioning of PJ ecosystems.

## Materials and methods

### Study sites

We examined PJ woodlands along two elevational gradients spanning ~300m on opposing sides of the San Francisco Peaks in northern Arizona (Fig 1 and S1 Table). We sampled eight sites, four per gradient, encompassing the highest, lowest, and mid-elevational ranges of pinyon (*Pinus edulis*) and single-seed juniper (*Juniperus monosperma*) co-occurrence. Other plant species within these PJ woodlands include Gambel oak (*Quercus gamebelii*), sagebrush (*Artemisia* sp.), rabbitbrush (*Chrysothamnus* sp.), bitterbrush (*Purshia* sp.), snakeweed (*Gutierrezia sarothrae*), many grasses and forbs (e.g., *Bouteloua gracilis*, *Draba* sp., *Erigeron divergens*, *Microsteris gracilis*, *Penstemon* sp., *Poa fendleriana*), and cacti (*Escobaria* sp., *Opuntia* sp.). The northeast gradient is located within the rain-shadow effect created by the San Francisco Peaks (highest point at 3851m elevation), and is therefore much drier than the northwestern gradient. We refer to these two gradients hereafter as the dry and wet gradients.

The relative proportions of pinyons and junipers changes across elevation [5, 6]; to describe tree composition differences among our sites we counted trees (greater than breast height) in five parallel 200m² plots at each site (Fig 2 and S1 Table). We found compositional trends typical for PJ woodlands along elevational gradients [3] with trees becoming more abundant with increasing elevations, junipers largely dominating low- to mid-elevations (i.e. 1911-2085m), and pinyons dominating the two highest elevation sites (2104m and 2200m). These two highest sites also contained small numbers of ponderosa pine (*Pinus ponderosae* Dougl. Ex. Laws.).

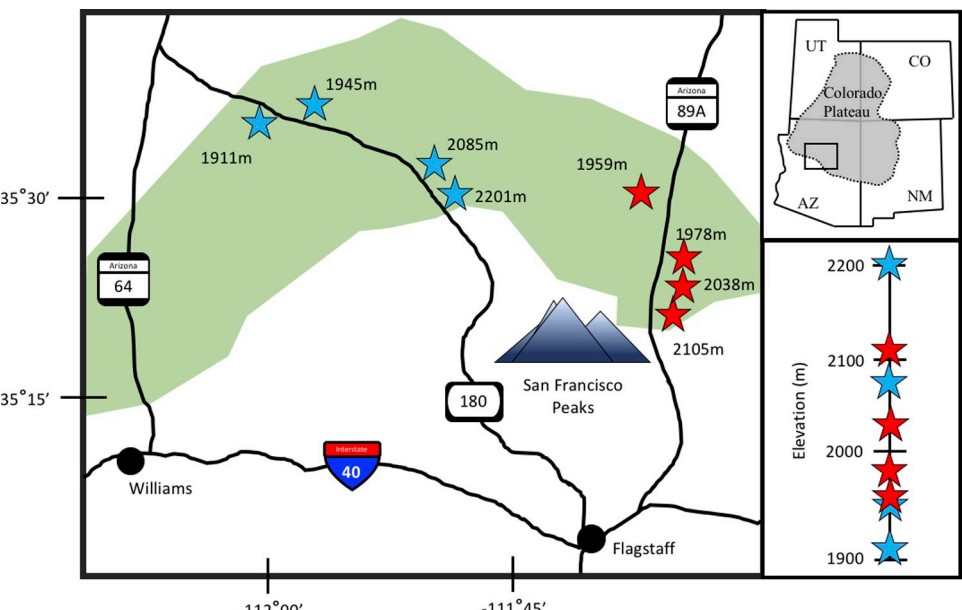

**Fig 1. Site locations.** Map of eight sites (represented by stars) on two elevational gradients in PJ woodland (green shading) in northern Arizona. The first gradient (red) is in the rain shadow of the San Francisco Peaks, while the second gradient (blue) receives higher precipitation.

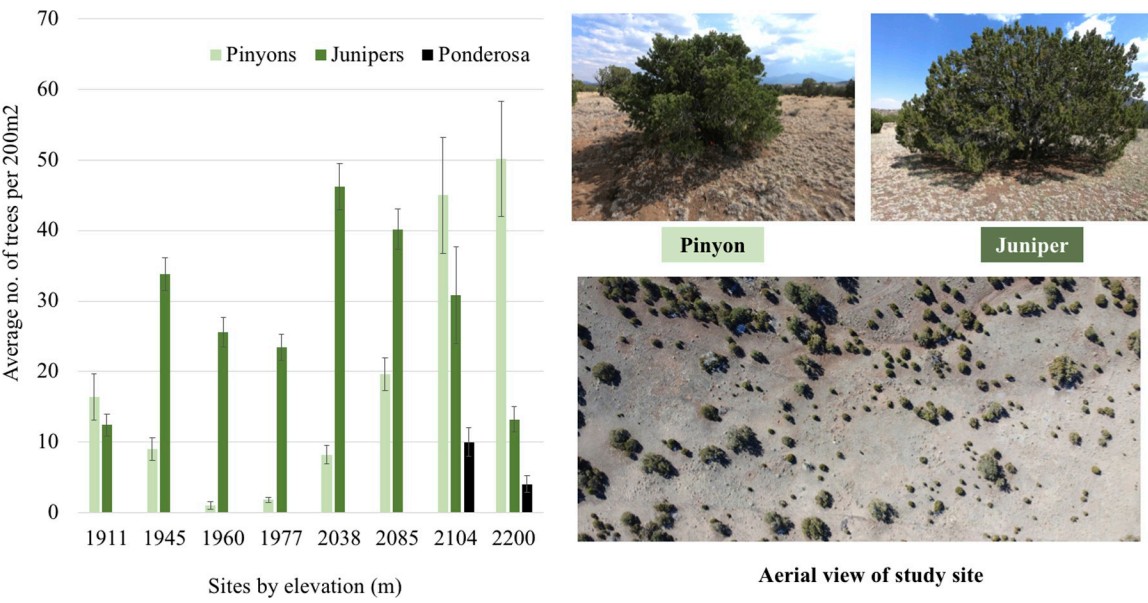

**Fig 2. Woodland composition of sites.** Tree composition of elevational sites averaged from five 200m$^2$ plots and photos of tree species and aerial (drone) view of "Blue Chute" study site (1945m).

While the proportion and number of trees changes among our sites, all these woodlands are characterized by generous spacing between trees with mature pinyons and junipers growing separately with wide sprawling canopies (Fig 2). These canopies commonly reach 5m in diameter, creating large areas of influence on the ground through shade and accumulated litter distinct to either pinyon or juniper. We sampled from these areas using the same three pairs of pinyon and juniper trees across five 7-day sampling periods spanning two years at each site. We purposefully chose large-canopied pinyons or junipers (>3m diameter canopy) that were not growing close (>5m) to the opposing species to avoid confounding effects. To ensure samples were representative of their respective tree species, we counted all trees within a 20m$^2$ diameter of sample locations (S1 Table), which showed that nearest tree neighbors were always the same tree species (or no other trees occurred), and within a 10m$^2$ diameter no opposing species occurred. We conducted our research on National Forest Service and private lands, with permission from Babbitt Ranches (https://www.babbittranches.com/) and project approval from the Southwestern Experimental Garden Array (https://sega.nau.edu/). Our study did not involve endangered or protected species.

**Seasonality and weather.** We sampled arthropod communities for one-week intervals five times, encompassing different seasons with vastly different weather patterns. Two sample periods occurred during dry summers (8–15 June 2018 and 27 June-4 July 2019) and one during a dry spring (7–14 April 2019); no precipitation fell within these sampling periods or during the month prior to sampling. Two other sample periods occurred during an above average monsoon season with large precipitation events during sampling (30 July -6 August 2018 and 2–9 Sept. 2018). We refer to the former three dates as dry season and the latter two as monsoon season.

**Climate and plant productivity measures.** We calculated climate measurements for each site using 30-year averages of annual total precipitation, average temperature, and average vapor pressure (S1 Table) with data extracted from PRISM Climate Group, Oregon State University (PRISM Climate Group, Oregon State University, http://prism.oregonstate.edu, created 10 Nov. 2019). These environmental factors contribute to understanding desiccation stress, a

key limit on arthropods in arid climates [33]. To estimate productivity, which is associated with resource availability, we used normalized difference vegetation index (NDVI, [35]) calculated from satellite imagery at the 250m resolution for each site and sampling period. NDVI is widely used, creating an index that ranges from zero to one with higher values indicating higher productivity [35]. PRISM data were downloaded at a spatial resolution of 800 meters and extracted using R.3.2.3 and the package raster version 2.5–2 based on observed latitude and longitude of each site.

**Arthropod sampling.** We used pitfall trapping to sample GDAs, with a trap dug well under the canopy of each tree as close to the trunk as possible. Pitfalls are good for sampling different GDA communities with equal intensity among treatments [36]. Traps were dug to ground level and opened during sample periods, with capped traps keeping locations constant in-between sampling. Pitfall traps consisted of a long borosilicate glass tube measuring 32 mm diameter and 200 mm length filled with 100ml of propylene glycol fitted within PVC sleeves with a rain cover ~3-4cm above the ground [27]. Some samples (12 of 240) were lost to flooding. We sorted pit trap samples and deposited voucher specimens at the Northern Arizona University Forest Entomology Collection, yielding richness and abundance measures for each trap. Taxa were identified and assigned functional groups with the assistance of taxonomic experts. We excluded adult Diptera and Lepidoptera as by-catch (i.e. not GDA). We digitally cataloged all reference species in Symbiota Collection of Arthropod Network (SCAN, http://scan-bugs.org) and http://bugguide.net (url numbers in S4 Table) online data portals.

## Data analysis

To investigate GDA composition in relation to categorical (i.e. tree species, date, and site) and continuous (i.e. elevation, climate, and NDVI) variables, we conducted complementary analyses on the GDA community as a whole, and separately on two dominant taxa groups (ants and beetles which together constituted 87% of individuals collected) using R.3.6.2 (R script: S1 File).

**Environmental variables.** To understand how climate and productivity changed across our gradients, we used simple linear regressions testing elevation against climate (temperature, precipitation, and vapor pressure) and NDVI variables. Temperature and vapor pressure were strongly correlated with elevation, while the variation between gradients caused precipitation to not correlate with elevation when all sites were considered together (Fig 3 and Table 1). NDVI was generally correlated with elevation, except during the dry April 2019 sample period (Fig 4 and Table 1). To analyze the effect of these environmental variables on GDA, we chose to test elevation (closely associated with temperature and vapor pressure), precipitation, and NDVI as separate variables, scaling them prior to analysis. These variables are not entirely independent, and we acknowledge it is difficult to fully separate their effects. However, we can infer their relative effects on GDAs by differences in patterns across sites.

**GDA abundance and richness.** To test whether richness and abundance of GDAs changed between pinyon and juniper trees, and along climatic gradients, we ran generalized linear mixed effect models (GLMM). GLMMs offer a flexible approach for testing multiple categorical and continuous variables, and are not subject to assumptions of sphericity. We examined abundance and species richness of the entire GDA community, two major insect taxa (ants and beetles), and remaining GDA groups (hereafter referred to as 'others') as our response variables. Because our response variables were count data, we ran each model with Poisson and negative binomial distributions. Using the Akaike information criterion (AIC, corroborated with AICc), we determined greater fit for the negative binomial distribution models and Poisson for GDA response variables (S2 Table). For all models, we treated scaled environmental variables (elevation, precipitation, and NDVI) and tree species as fixed effects while date and site were treated

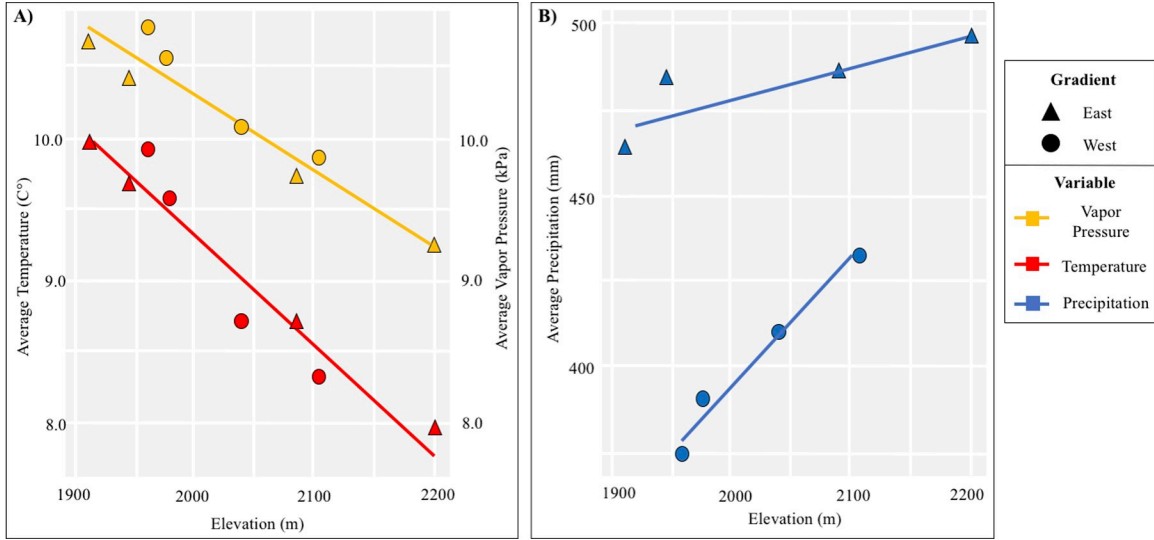

**Fig 3. Temperature, vapor pressure, and precipitation along gradients.** Averages of climate variables along elevational gradients (shapes). (A) Average temperature (red) and vapor pressure (yellow) decrease with elevation. (B) Precipitation (blue) is lower on the east gradient and increasing pattern with elevation is only evident when gradients are separated.

as crossed random effects. We used Wald $\chi^2$ tests to exclude non-significant predictor variables and AIC score comparisons among models with all possible variable combinations to select final models (S2 Table). We checked model performance graphically via diagnostic plots [37]. We performed GLMMs with R packages lme4 [38] and arm [39].

**GDA composition.**  We assessed individual GDAs and whole community composition responses to tree species, climate, and NDVI variables across sites and dates with the function *manyglm* from R package mvabund [40]. In this function, each taxon is fitted with individual generalized linear models (GLM) providing both taxa-level and global estimates of significance while controlling for multiple testing. Variance in abundance was greater than the mean for most taxa, therefore abundance of taxa $j$ in sample $i$ was modeled as negative binomial. Models included tree species, site, date, elevation, precipitation, and NDVI as predictor variables. We report global (entire GDA community) significance for each explanatory variable, and specific relationships of continuous variables with taxa with more than five observations. For categorical variables, we used indicator analyses for taxa-level results. We conducted indicator species analysis using R packages labdsv [41] and indicspecies [42] packages.

**Table 1. Elevation and climate variable correlations.**

|  |  | $r$ | adjust-$R^2$ | p-value | $F_{1,6}$ |
|---|---|---|---|---|---|
| Elevation | Av. Temperature | -0.97 | 0.93 | <0.0001 | 88.98 |
|  | Av. Vapor Pressure | -0.96 | 0.91 | 0.0002 | 67.65 |
|  | Av. Precipitation | 0.38 | 0.01 | 0.352 | 1.02 |
|  | NDVI June 2018 | 0.87 | 0.71 | 0.005 | 18.12 |
|  | NDVI August 2018 | 0.95 | 0.9 | 0.0002 | 60.38 |
|  | NDVI Sept. 2018 | 0.93 | 0.84 | 0.0008 | 39.02 |
|  | NDVI April 2019 | 0.25 | 0.01 | 0.554 | 0.39 |
|  | NDVI July 2019 | 0.94 | 0.87 | 0.0005 | 46.35 |

Correlations of environmental variables and productivity (NDVI: normalized difference vegetation index) with elevation across eight sites.

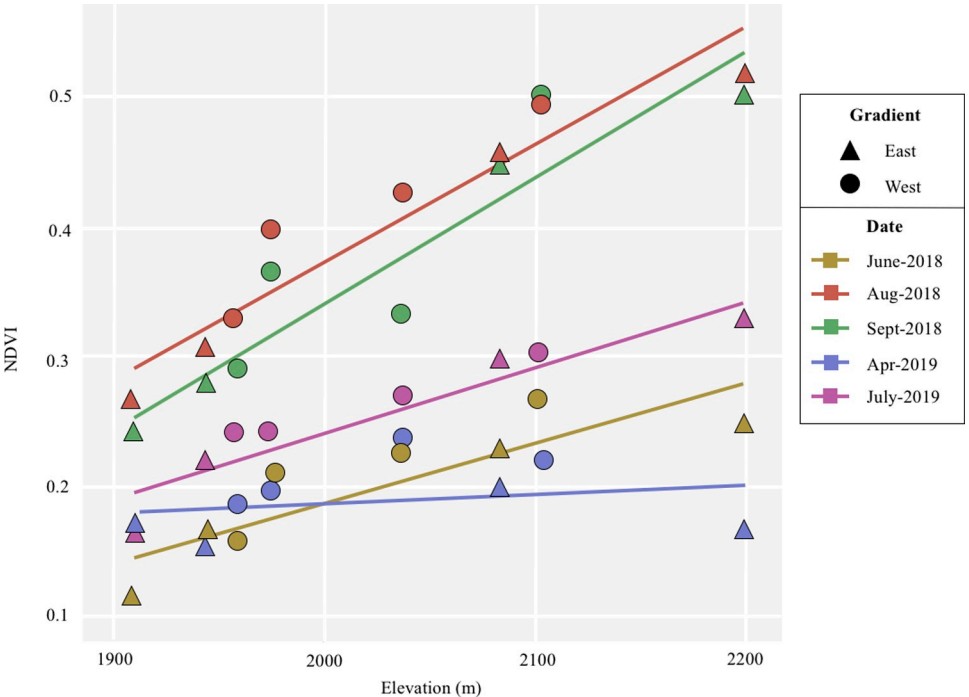

**Fig 4. NDVI along gradients.** Normalized difference vegetation index (NDVI) measurements from five different dates (color) along two elevational gradients (shape). NDVI is higher during monsoon sampling periods (August and September 2018) than dry sampling periods. For all dates except April 2019, NDVI has a strong positive correlation with elevation.

**Ants and beetles.** To determine whether patterns differed between our two most dominant taxa, we analyzed ants and beetles separately through comparison of dissimilarity matrices from averaged (across site/date/tree) data. Beetle data required a square root transformation, following which we constructed similarity matrices with Bray-Curtis similarity coefficients. Ants showed extreme variation, with many high-abundance outliers likely caused by ant nest proximity. We therefore analyzed ant data on an incidence basis with Jaccard similarity coefficients for similarity matrices. We visualized results via multi-dimensional scaling (MDS) plots. We used goodness of fit, Shepard diagrams, and stress to test satisfactory fit of ordination (S2 File). To test if ants or beetles differed by date, site, or tree species, we used permutational analysis of variance (PERMANOVA, permutations = 9999) including all factors with significant differences followed by pairwise comparisons. We checked PERMANOVAs and pairwise results against analysis of similarity (ANOSIM) and multiple response permutation procedure (MRPP) (S3 Table); all analyses agreed, we report PERMANOVA results. We fitted environmental variables to ordinations and permutations to test significance (permutations = 999). All analyses were done using R packages vegan [43] and ecodist [44].

## Results

Over two years of sampling, 12,365 GDAs were captured with 115 taxa identified (S3 Table). Species accumulation curves show an asymptote for sampling both juniper and pinyon trees at each site (S1 Fig). Abundance was dominated by ants (76.0% of individuals) followed by beetles (11.0%), spiders (3.9%), a morphospecies of slender springtail (3.8%), orthopterans (1.2%), mites (1.2%), non-ant hymenopterans (1.1%), hemipterans (0.8%) and others (2.0%) (S4 Table). Mites, parasitic wasps, and pseudo-scorpions (<2% of individuals) were unable to be

morphotyped and were identified to order only, with other arachnids (e.g. spiders) identified to family (<4% of specimens). All other arthropods (96.0% of specimens) were identified to species (81.2% of specimens) or morphospecies (14.8% of specimens). The most diverse GDA were beetles (52/115 taxa) followed by ants (21/115). We use the terms richness and diversity for referring to these mixed taxonomic levels. Functionally, GDA were mostly omnivores (76.0% of specimens, all ants), followed by detritivores (13.0% of specimens, mostly beetles in the family Tenebrionidae), predators (6.3%, mostly arachnids), fungivores (1.8%, mostly beetles in the family Nitidulidae), and herbivores (1.2%, mostly beetles, S4 Table).

### Insect community in pinyon vs juniper

Contrary to our hypothesis, we detected no significant differences among any GDA community metric or taxa analyses between pinyon and juniper on any GDA community metric or taxa in any analysis, except a single morpho-type of *Leiodes* (family Leiodidae) beetle which was three times more abundant in and a significant indicator (IV = 24.3, $p$ = 0.023) of junipers. Besides 33 rare taxa (<6 individuals collected), pinyon and juniper shared all taxa at nearly equal abundances.

### Gradient and season

Multivariate GLM showed GDA communities differed significantly among sites (Deviance [Dev] = 1952, Pr (>Dev) = 0.001) and dates (Deviance [Dev] = 1634, Pr (>Dev) = 0.001). Twelve taxa were indicative of one site and 23 taxa were indicative of one date (S5 Table). For dates, pairwise differences in ants and beetles were largely between dry and monsoon dates with most dates within-seasons being similar. For sites, pairwise differences were largely between east and west gradient sites with most within-gradient sites similar (S6 Table).

### Climate and productivity

GDA richness was positively correlated with precipitation (Fig 5 and Table 2). GDA compositional differences were linked to both precipitation (Deviance [Dev] = 405.9, $p$ = 0.001) and

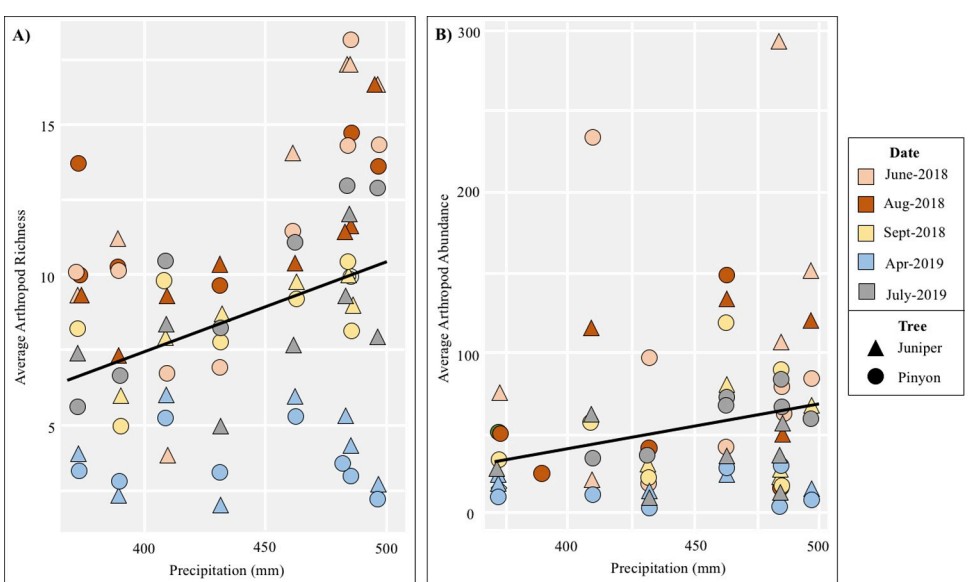

**Fig 5. GDA richness and abundance along precipitation gradients.** (A) GDA richness and (B) abundance increase with precipitation significantly when date and site are random effects (Table 2).

elevation (Deviance [Dev] = 156.8, $p$ = 0.001) and marginally to NDVI (Deviance [Dev] = 93, $p$ = 0.056). Nine taxa increased significantly with elevation, two taxa (*Forelius pruinosus* and *Monomorium cyaneum*) decreased with elevation, and one taxa (Lycosidae) increased with precipitation (S7 Table).

### Ant and beetle differences

Ants and beetles differed in relationships with climate variables and NDVI. Ant richness increased with elevation and precipitation, but ant abundance showed no relationships with these variables (Table 2). Ordinations of ant composition showed significant correlations with NDVI (($R^2$ = 0.163, $p$ = 0.004), Fig 6 and S8 Table). Beetle abundance and beetle richness increased with precipitation (Table 2), and ordinations of beetle composition showed significant correlations with elevation ($R^2$ = 0.211, $p$ = 0.002), precipitation ($R^2$ = 0.191, $p$ = 0.003) and NDVI ($R^2$ = 0.085, $p$ = 0.034) (Fig 6 and S8 Table). Other GDA richness and abundance increased with precipitation (Table 2).

## Discussion

### Tree species effect

Pinyon-juniper (PJ) woodlands provide habitat for a wide range of arthropods in one of the driest and hottest forest types in western North America. Pinyon and juniper trees may have unique roles in shaping and maintaining insect biodiversity. Here, we tested if ground dwelling arthropods (GDA) specialize in either pinyon or juniper niches, assuming differences between litter composition or habitat structure would create differences in local GDA communities. Despite considerable differences in tree chemistry [8], wood properties [9, 10], litter and soil characteristics [11, 12] and canopy architecture [13] of pinyon and juniper, we found only one GDA that showed affinity to one of these tree species. Our results suggest that most GDAs in PJ woodlands utilize both tree species, but our patterns should be cautiously extrapolated to PJ woodlands in other regions that contain different juniper species. A limitation of ours and most studies examining the full range of GDA diversity is taxonomic clarity for certain groups. We identified 81 percent of specimens to species but the rest were assigned to either morphospecies or higher taxa levels, which may obscure patterns for these latter groups.

**Table 2. GLMM models for GDA richness and abundance.**

| Response variable | Predictor variable(s) | Estimate | SE | Z | *P* |
|---|---|---|---|---|---|
| GDA Abundance | *No significant model* | | | | |
| GDA Richness | Precipitation | 0.18 | 0.04 | 4.45 | <0.001 |
| Ant Abundance | *No significant model* | | | | |
| Ant Richness | Precipitation | 0.16 | 0.04 | 3.69 | <0.001 |
| | Elevation | -0.15 | 0.07 | -2.067 | 0.038 |
| Beetle Abundance | Precipitation | 0.4 | 0.12 | 3.18 | <0.001 |
| Beetle Richness | Precipitation | 0.2 | 0.07 | 3.07 | 0.002 |
| Other GDAs Abundance | Precipitation | 0.177 | 0.049 | 3.62 | <0.001 |
| Other GDAs Richness | Precipitation | 0.344 | 0.126 | 2.74 | 0.006 |

Final models with significant predictor variables. Comparisons included tree species, elevation, precipitation, and NDVI with site and date as crossed random effects (S2 Table). Models were run on richness and abundance of A) all arthropods, B) only ants, C) only beetles, and D) other GDAs. Intercept estimates, standard errors (SE), Z-values (Z), and p-values (P) are given for each variable.

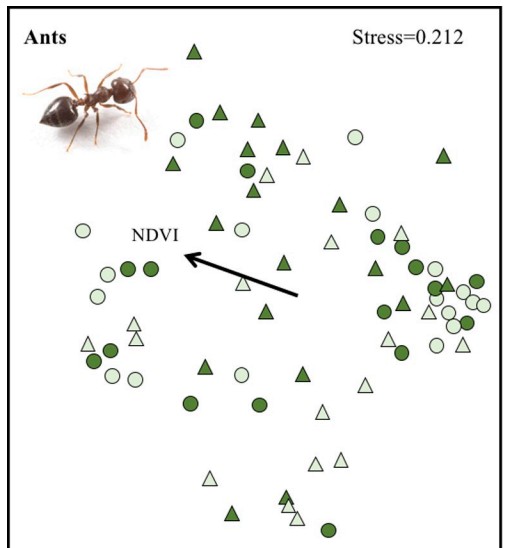
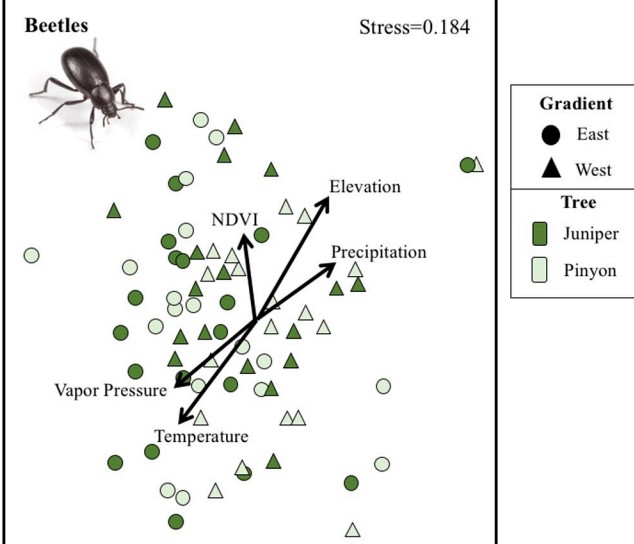

**Fig 6. Multi-dimensional scaling plots of PJ woodland ant (left panel) and beetle (right panel) communities using the Bray-Curtis similarity coefficient.** Significant correlations of environmental variables are mapped onto ordinations. Sites show significant group differences, with most pairwise differences (S6 Table) among gradients (color). Tree species (shape) shows no significant differences (PERMANOVA, $F = 1.42$, $p = 0.196$).

The use of tree microhabitat by GDAs may be either specific or random. Our study suggests the latter for PJ woodlands, yet, there are many instances where GDAs are sensitive to changes in vegetation structure [45–49]. Canopy-closure creates microhabitats which host different GDA communities than open areas [50], including in PJ woodlands [26–28]. So, while most GDAs may not be affected by tree composition, some may be confined to canopied habitat with open habitats acting as barriers [46]. How isolated a tree is may affect what GDAs occur under it. These effects may be exacerbated in species such as ants, which typically stay close to nests [51]. Our trees varied in their level of isolation, as the densities of trees varied across sites (Fig 2). PJ woodlands typically have wide spacing between trees and our results may not be applicable to more mesic forest types with unbroken canopies.

Foliar and wood-infesting arthropod communities commonly differ among tree species [7, 17] but reasons exist to doubt whether tree species create unique GDA communities. Increasing tree diversity does not seem to increase GDA diversity [52, 53]. Many GDAs are generalists; three-quarters of specimens in our study were omnivorous ants and the remaining were largely detritivores or predators. Detritivores pull nutrients from litter and microbial turfs decomposing litter [54]. While litter quality does differ under tree species, many plant defensive compounds break down over time, meaning less specialization is required to ingest litter or microbes consuming litter, versus plant tissues [55]. We found many generalist detritivores such as darkling beetles, which may not be heavily influenced by the source of litter (e.g. [52]. With GDA prey not different among trees, there may be little pressure for predators to differ (e.g. [53]).

Only a small minority of specialist taxa are likely tied to tree species. The majority of GDA in our study were found under both tree species in surprisingly equal abundance, except one beetle (*Leiodes* sp.) indicative for junipers that likely specializes in subterranean fungi [56]. Pinyon and juniper host different mycorrhizal mutualists [57] potentially explaining the association. Specialized relationships like these, and therefore community differences among tree species, are more likely to arise with herbivorous or fungivorous relationships. Future studies

should explore foliar and wood-infesting arthropod communities of pinyon and juniper, which are more likely to have specialist species driving trends.

## Climate and productivity effects

Our elevational gradient, which only encompassed PJ woodlands, is short compared to most elevational studies, yet we still found strong climate-driven trends. Typically, temperature is posited to drive arthropod community dynamics along elevational gradients [33, 58]. Increases in temperature (and thus decreases in elevation) are associated with increased arthropod abundance and richness. Temperature at our sites showed strong correlations with beetle composition, and correlations of some taxa with elevation suggest temperature strongly affects them as well. However, elevation was not a strong predictor of GDA richness or abundance. While we cannot fully disentangle the roles of covarying environmental factors (i.e. temperature, vapor pressure, precipitation, and productivity), the comparatively strong correlations of precipitation with GDA richness and abundance, along with differences between dry/wet gradients and dry/monsoon dates, highlight the strong role of water in our system.

Arid systems commonly have "flipped" elevational patterns, where richness and abundance increase with elevation due to lack of water at low elevations [58, 59]. Our results suggest PJ woodlands fit this pattern. In arid systems like Arizona, limited water at low elevations puts those communities at immediate risk from increased severity and frequency of droughts. Across most PJ woodlands, average annual temperatures have risen over 1.0˚C since 1950, warming at a much faster rate than most of the continent [60]. Further warming of 1.5–2˚C and increased droughts by 2050 are predicted for the region [61]. This is expected to amplify tree mortality and shift forest compositions [5]. Recent droughts and drought-induced pests have resulted in high mortality of pinyon [3, 6, 62]. Low-elevation PJ woodlands are often the first to succumb to drought [3], and GDAs may be equally or even more drought susceptible. At higher elevations, PJ woodlands are replacing ponderosa pine forests [6, 62]. GDAs may follow this progression or precede it. Our results suggest the latter, since GDAs were not dependent on tree species.

Our results clarify GDAs vulnerabilities to environmental changes in PJ woodlands. Differences among sample dates appear to be related to seasonal precipitation patterns while differences among sites likely relate to long-term precipitation patterns. While rarely compared, climate does seem to have a stronger direct effect on certain GDA communities than primary productivity [58, 63, 64], suggesting that GDA communities respond directly to changes in climate, rather than changes to vegetation and habitat structure.

## Taxa differences

Environmental patterns were not concordant among GDA community metrics (richness, abundance, and composition) and taxa (ants and beetles). Ants were extremely abundant and varied extensively, a common phenomenon for GDA surveys caused by nest and foraging trail proximity [65]. Non-results of ant (and overall GDA) abundance may be due to sampling method, as pit traps may not accurately sample ant abundance, but significant patterns were still found with ant richness. Beetles made up roughly half the diversity, showing strong abundance and richness patterns. Both ants and beetles differed in their responses to elevation, precipitation, and NDVI. These relationships changed whether groups were measured by richness, abundance, and composition. Furthermore, taxa responded differently when analyzed individually. These nuances highlight the challenge of understanding GDA communities, and studies must examine a large spectrum of measurements within a diverse range of taxa to be comprehensive.

## Conclusion

GDAs are important for nutrient-cycling and support higher trophic levels; understanding their relationships to climate and habitat may be critical for effective management of PJ woodlands. Currently, most management focuses on woodland reduction to mitigate conifer encroachment on ungulate habitat or reduce wildfire risks [66]. Bombaci & Pejchar [67] highlighted the potential negatives of this strategy for wildlife and our lack of knowledge on invertebrates in PJ ecosystems. GDA communities are sensitive to changes in habitat following drought-induced [27] and fire-induced [28] mortality of pinyon. Reduction of PJ woodland, whether through anthropomorphic or environmental means, likely has consequences for GDA biodiversity and ecosystem functions. With the threat of intensified droughts and increased temperatures, we underscore the need to establish more baseline data of arthropod communities in PJ woodlands to better understand and conserve these ecosystems.

## Supporting information

**S1 Table. Site locations and characteristics.** Includes site coordinates, elevations, and estimates of climate variables, primary productivity, 200m$^2$ plots for estimating tree species composition at site level, and 10m$^2$ radius plots for estimating tree composition at sample level.
(XLSX)

**S2 Table. GLMM model selection showing the two-model selection approach with AIC comparisons.** All models included date and site as random effects, only predictor variables are included here.
(XLSX)

**S3 Table. PERMANOVA, ANOSIM, and MRPP results of ant and beetle communities across tree species, sites, and dates.** All analyses agreed.
(XLSX)

**S4 Table. Arthropod data showing all taxa.** Raw data and taxonomic identifications of arthropods. Diptera and Lepidoptera were not used in analysis.
(XLSX)

**S5 Table. Indicator analysis for tree species, site, and date.**
(XLSX)

**S6 Table. Pairwise site and date differences of ant and beetle community composition.** Site differences were largely between sites on different gradients (brown = dry gradient, green = wet gradient) and date differences were largely between dates in different seasons (dry = red, monsoon = blue).
(XLSX)

**S7 Table. GLM results for each taxon with environmental variables via manyGLM function.** Each model included predictors of elevation, precipitation, and NDVI. Correlations show direction of effect.
(XLSX)

**S8 Table. Correlations of environmental variables with ant and beetle ordinations.**
(XLSX)

**S1 File. Zip files with R script and output, and csv datasheets for all analyses.**
(ZIP)

**S2 File. Table of ordination fit statistics (stress and goodness of fit) and shepard diagrams.**
(DOCX)

**S1 Fig. Species accumulation curves.** Number of unique taxa accumulated during sampling for both tree species (pinyon and juniper) at elevational sites.
(TIF)

## Acknowledgments

We thank Michael Remke, Rebecca Hirsch, and Sneha Vissa for helping with data collection. We also thank Neil Cobb, Babbitt Ranches, the Southwestern Experimental Garden Array (SEGA, established under National Science Foundation award #1126840 and Field Stations and Marine Labs Grant #152253) for providing site access. This manuscript benefited from the input of many arthropod taxonomists who helped identify taxa including Dr. Peter Messer (ground beetles), Dr. Gary Alpert (ants), Dr. Matthew Prebus (acorn ants), Nick Fensler (spider wasps), Dr. Bill Warner (scarab and clown beetles), Dr. Ainsely Seago (Leiodidae), Dr. Donald Chandler (Anthicidae), Dr. Robert Anderson (weevils), Dr. Charlene Wood (Latridiidae), Gareth Powell (Nitidulidae), Dr. Blaine Mathison (click beetles), Anthony Cognato (bark beetles), and Vassili Belov.

## Author Contributions

**Conceptualization:** Derek Andrew Uhey, Richard William Hofstetter.

**Data curation:** Derek Andrew Uhey, Hannah Lee Riskas, Aaron Dennis Smith, Richard William Hofstetter.

**Formal analysis:** Derek Andrew Uhey, Hannah Lee Riskas, Aaron Dennis Smith.

**Investigation:** Derek Andrew Uhey, Hannah Lee Riskas.

**Methodology:** Derek Andrew Uhey, Hannah Lee Riskas, Richard William Hofstetter.

**Project administration:** Derek Andrew Uhey.

**Resources:** Derek Andrew Uhey, Richard William Hofstetter.

**Software:** Derek Andrew Uhey.

**Supervision:** Derek Andrew Uhey.

**Validation:** Derek Andrew Uhey.

**Visualization:** Derek Andrew Uhey.

**Writing – original draft:** Derek Andrew Uhey, Hannah Lee Riskas, Richard William Hofstetter.

**Writing – review & editing:** Derek Andrew Uhey, Hannah Lee Riskas, Aaron Dennis Smith, Richard William Hofstetter.

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
