## [Decision Letter · Decision Letter 0]

19 May 2020

PONE-D-20-11956

Ground-dwelling Arthropods of Pinyon-Juniper Woodlands: Arthropod Community Patterns are Driven by Climate and Overall Plant Productivity, not Host Tree Species

PLOS ONE

Dear Mr. Uhey,

Thank you for submitting your manuscript to PLOS ONE. After careful consideration, we feel that it has merit but does not fully meet PLOS ONE’s publication criteria as it currently stands. Therefore, we invite you to submit a revised version of the manuscript that addresses the points raised during the review process.

I appreciate the amount of work put in by the authors, on both the analyses and the manuscript. In particular, I found the Introduction section to be well-written, with clear explanations of the authors' objectives and expectations. Nevertheless, I think there are still some aspects to confront before the manuscript may be suitable for publication. The reviewers have provided detailed comments that should serve as an excellent guide during the revision process. Foremost, I would focus on addressing the first major concern raised by each reviewer (Reviewer 1 about sampling methods and power, Reviewer 2 regarding level of taxonomic identification), but it is important to consider and respond to all of their concerns. I also agree with the reviewers that the Discussion section should be improved and expanded. They have provided several potential jumping-off points that may be worth further examination.

We would appreciate receiving your revised manuscript by Jul 03 2020 11:59PM. To enhance the reproducibility of your results, we recommend that if applicable you deposit your laboratory protocols in protocols.io, where a protocol can be assigned its own identifier (DOI) such that it can be cited independently in the future. For instructions see: http://journals.plos.org/plosone/s/submission-guidelines#loc-laboratory-protocols

We look forward to receiving your revised manuscript.

Kind regards,

Frank H. Koch, PhD

Academic Editor

PLOS ONE

Additional Editor Comments (if provided):

In Table S5, there are cases where no arthropods were reported for a given tree on a given sampling date. For example, there is no June 2018 record for tree J20 or J24.  I assume these were the samples that were lost to flooding (line 135). I suggest inserting a row for each of these samples, indicating that they were lost to flooding and so there are no data to report.

2. We note that Figure 1 in your submission contains map images which may be copyrighted. All PLOS content is published under the Creative Commons Attribution License (CC BY 4.0), which means that the manuscript, images, and Supporting Information files will be freely available online, and any third party is permitted to access, download, copy, distribute, and use these materials in any way, even commercially, with proper attribution. For these reasons, we cannot publish previously copyrighted maps or satellite images created using proprietary data, such as Google software (Google Maps, Street View, and Earth). For more information, see our copyright guidelines: http://journals.plos.org/plosone/s/licenses-and-copyright.

Reviewers' comments:

Reviewer's Responses to Questions

**Comments to the Author**

1. Is the manuscript technically sound, and do the data support the conclusions?

Reviewer #1: Partly

Reviewer #2: Partly

2. Has the statistical analysis been performed appropriately and rigorously? 

Reviewer #1: Yes

Reviewer #2: No

3. Have the authors made all data underlying the findings in their manuscript fully available?

Reviewer #1: Yes

Reviewer #2: Yes

4. Is the manuscript presented in an intelligible fashion and written in standard English?

Reviewer #1: Yes

Reviewer #2: Yes

5. Review Comments to the Author

Reviewer #1: In this manuscript, Uhey et al. examine the role of tree types, climate and productivity on the richness and abundance patterns of ground dwelling arthropods in Pinyon-Juniper woodlands. They sampled ground arthropods using pitfall traps at several sites underneath Pinyon and Juniper trees at several time points and use generalized linear mixed models and other methods to address their question. I think they posed a very interesting question. Often when we think about arthropod-plant interactions, we think of foliage-dwelling arthropods and not ground-dwelling arthropods. The authors did a really good job of explaining how ground dwelling arthropod communities could be affected by tree types in the introduction and they also collected a lot of arthropods. However, I was not entirely convinced by their sampling methods, at least as they are currently written. I think the results section needs more details as well. They used a whole host of statistical methods but do not report the results of these in sufficient detail in the manuscript. Finally, I think their discussion needs to be developed further. They left out some important papers that they should have cited and do not address the limitations of their current work or point to any future directions.

Major comments:

1. I am concerned about the sampling method used in the paper and whether the authors even had enough power to discern differences in the ground dwelling arthropod communities harbored by Pinyon vs Juniper trees. They only sampled under three Pinyon and three Juniper trees at each site, which I am not sure gives them enough power to be able to discern any differences between arthropod communities under those trees. They also chose isolated trees for sampling (also need to clarify how isolated these isolated trees were, i.e. how far from the nearest tree?). However, there might be confounding factors that lead to these trees being isolated that affect the arthropod community. I think a better study design would have been to record the density of juniper and pinyon trees in say 10 m2 radius around each pitfall trap and look at how variation in the density of juniper and pinyon trees in an area affects arthropod diversity. If they sampled an isolated juniper tree that was in an area dominated by Pinyon trees, then are they sampling ground-dwelling arthropods that might prefer juniper tree litter? I am not sure. I think they need to add more details about the relative abundance of Pinyon and Juniper trees around their pitfall traps. If all the sites had a similar density of Pinyon and Juniper trees around the pitfall traps, then I do not think they are able to test the effects of tree type at all.

2. I think it would have been interesting if the authors had also sampled foliage dwelling arthropods. That would have set up a cool comparison of how diversity patterns of foliage dwelling versus ground dwelling arthropods are affected by tree types, climate, and productivity. This might have also allayed some concerns about sampling adequacy if they had noticed an effect of tree types on foliage dwelling arthropods but not ground dwelling ones. Unless the authors already have these data, I know it might not be feasible to go and collect foliage dwelling arthropods now, but this is something they could include in the future directions section of the discussion.

3. In the first sentence of the study sites section in Methods, the authors say that they sampled along two elevational gradients spanning ~1000m. However, when I look at figure 1, it looks like all their sites are between 1900m-2200m elevation. If that is indeed the case, then the elevational gradient is just 300m. I think it is important for the authors to clarify this. I would also be careful when citing literature on elevational gradients and comparing their results to other studies on elevational gradients. This study only captures a subset of the entire gradient, so any patterns seen or not seen with regards to elevation need to be interpreted with that in mind. I am not suggesting that they should have sampled a longer gradient; given their research question with regards to Pinyon-Juniper woodlands, I think the elevational distribution of their study sites make a lot of sense. I am just suggesting a more nuanced explanation of results in terms of elevation.

4. I am unclear about the specification of the GLMMs. Date and site were used as random effects, but were they entered as two separate random effects or were they treated as crossed random effects? As far as I understand their sampling design, they should have been treated as crossed random effects. Did the authors run model diagnostics to make sure that the model specification was okay? Also, why the use of AIC instead of AICc which corrects for small sample size? Lastly, I think they should include their GLMM table in the main paper instead of putting it in the supplement. At least the table for all arthropods. The model should include standard errors in addition to the slope values. That would help understand if they had enough statistical power. I would also encourage the authors to include their R script as a supplement.

5. I think the authors need to look through the literature more carefully and make sure to cite relevant studies. For example, I was wondering if there are other studies that show an effect of tree types on ground dwelling arthropods and a quick search led me to this paper: https://doi.org/10.1111/j.0030-1299.2008.15972.x I think this paper should be cited in this study. I would advise the authors to look through the literature once again and cite other relevant studies, it will make the discussion section much stronger. Also, like I said earlier, discussing the limitations of the study and what future work should be done in this field of research should be included in the discussion.

Minor comments:

I am personally not a huge fan of acronyms within papers unless they are already universally used. So, I think it is better to just say ground dwelling arthropods instead of GDAs and Pinyon-Juniper woodlands instead of PJs. I think it makes for easier reading, but it is up to you.

Line 39: need a comma between “tree specialization hypothesis, a specific version”

Line 40: “dominant” not “dominate”

Line 62: “management of these ecosystems”

Line 116: What was the source of this climate data? Did they have weather stations or is it satellite data from Worldclim? Please describe.

Line 124: Spatial resolution of 800m seems pretty low given the proximity of the sites that were sampled. I thought there was higher resolution data available for US, but I am not sure. MODIS seems to have data at 250m resolution(https://modis.gsfc.nasa.gov/data/dataprod/mod13.php) and has EVI in addition to NDVI which has been recommended in some recent work. I do not have expertise in this domain, but I would encourage the authors to look a little bit more into this.

Line 170: “Fig 3”, not “Fig 2”

Line 177: There is a typo, it should be “Linear” not “Liner”

Line 204: Please clarify what you mean when you say ant data was relativized by maximum.

Figure 3: This figure might not be readable to people with red-green colorblindness. Please check and try to avoid the use of red and green together in a figure. This is a really useful resource to help pick colors that are colorblind friendly: https://colorbrewer2.org/#type=sequential&scheme=BuGn&n=3

Reviewer #2: In this study, the authors compare communities of ground dwelling arthropods in pinyon-juniper woodlants using specimens from pitfall traps that were repeatedly sampled over the course of two years. With an emphasis on ants and beetles, they ask whether there are distinct species assemblages based on tree type, and also evaluate the effects of climate and productivity as measured by NDVI. They find variable effects of precipitation, NDVI, and elevation depending on the taxon and metric. While I think the idea is interesting, I have three main concerns:

1. Mixed genera, morphospecies, and species.

The authors do not identify all taxa, instead frequently using morphospecies or genera. This is common and completely understandable. However, the grouping is rather uneven, such that the analyses are performed on communities composed of a rather haphazard mixture of species, morphospecies, and genera. This makes me rather skeptical of the results. For example, several speciose genera (e.g., Crematogaster) are left as genera ('Crematogaster sp.'), while other smaller genera are identified to species (e.g., Forelius). I recognize that these genera are very difficult to identify, but I do not think it is valid to treat 'Crematogaster sp' as taxonomically equivalent to Forelius mccooki in terms of community composition and dissimilarity. This applies to other arthropods as well. For example, 'Melanotus similis' is treated as a separate taxon compared to 'Melanotus sp', but most specimens are identified only to genus. I expect that the taxa identified to species may have an outsized weight on the analyses, despite representing a minority of the actual species, while genera like Crematogaster sp. or Formica sp. may be obscuring differences between the communities since many species are treated as one. Obviously it would be best to identify all specimens to species (or to morphospecies representing a few likely species for those that may not be separable). Otherwise, identifying only to genera might be a better approach.

2. Treatment of ant worker abundances

The authors acknowledge that abundances of ant workers in their samples were biased by proximity of some pitfalls to ant nests, and they accounted for this by relativizing worker abundances by the maximum. However, this does not adequately or correctly account for the relevant peculiarities of ant life histories. In particular, ant foraging behavior varies by species, and some species are more likely than others to form foraging trails (Lanan 2014) that might lead to large numbers of ants from the same colony being captured in one trap (Bestelmeyer et al 2000). With ants in pitfall traps, it is unfortunately not possible to use the observed worker abundances as reliable estimates of either local worker abundance or local colony density. There are incidence-based metrics available that could be useful. A simple alternative would be to use a presence-based metric like Jaccard dissimilarity rather than Bray-Curtis (analyses in lines 206–207).

3. Discussion topics

The discussion should also be expanded. For example, it would be good to discuss the spatial scales involved in the context of tree specificity. Ground-dwelling arthropods generally move quite a bit, so the assumption you're making is that abundance in pitfall traps is correlated with 'time spent in that microhabitat'. Isolated trees were also intentionally selected for sampling to avoid mixed canopies of pinyon and juniper. Would this affect the observed community? What if the taxa that are specialized to tree type are also less likely to travel across larger stretches of open canopy? Would you expect differences between ants and beetles, given that ants are central place foragers with a (generally) stationary nest, unlike beetles?

As just an additional comment, this dataset seems to be very well-suited to using an occupancy model to account for species that were present but happened to not be sampled, with interesting combinations across aspect, elevation, and tree type. I recognize this is not the focus of this study, but it could be an interesting avenue for the authors to explore in the future.

References:

Bestelmeyer, B. T., Agosti, D., Alonso, L. E., Brandão, R. F., Brown, W. L. J., Delabie, J. H. C., & Silvestre, R. (2000). Field Techniques for the Study of Ground-Dwelling Ants: An Overview, Description, and Evaluation. In D. Agosti, J. D. Majer, L. E. Alonso, & T. R. Schultz (Eds.), Ants: Standard Methods For Measuring and Monitoring Biodiversity (pp. 122–144). http://antbase.org/ants/publications/20339/20339.pdf

Lanan, M. (2014). Spatiotemporal resource distribution and foraging strategies of ants (Hymenoptera: Formicidae). Myrmecological News, 20, 53–70.

Minor items

27–28: briefly summarise these points here

40: "dominant"

46–47: Doesn't the preceding sentence contradict this claim? If there are few studies which are largely limited to birds, how can you state that the shifting composition is changing animal communities?

48: maybe something like "Arthropods often closely associate with vegetation types"

49: specify 'plant' or 'tree' genotypes

56: Add "The" to the start of the sentence to make it a bit more clear

60–62: This sentence is rather vague. Could you tie it to the previous sentence more, specify how preservation and management could be improved, and/or add an example with a reference?

89: 'bitterbrush'

197–198: RStudio is just the IDE / interface - I think only the R version is necessary here.

221: Specify which Supp Table

332: I think here (and throughout) it would be good to qualify 'diversity' and 'richness', as readers will generally assume these are at the species level.

Fig. 4: I think it would be clearer to show the dates as different colors and the trees as different shapes

6. PLOS authors have the option to publish the peer review history of their article (what does this mean?). If published, this will include your full peer review and any attached files.

Reviewer #1: No

Reviewer #2: No

---

## [Author Response · Author response to Decision Letter 0]

1 Jul 2020

Dear Editor,

 Please find below our responses to reviewer comments for our manuscript, “Ground-dwelling Arthropods of Pinyon-Juniper Woodlands: Arthropod Community Patterns are Driven by Climate and Overall Plant Productivity, not Host Tree Species”, catalog number PONE-D-20-11956. We have made extensive revisions and believe our manuscript has greatly benefited from reviewer feedback. All authors have approved of changes.

Thank you for your time and consideration,

Derek Uhey, Corresponding Author

Northern Arizona University

College of Engineering, Forestry and Natural Sciences

200 E. Pine Knoll Dr.

Flagstaff, AZ 86011 USA

Ph: 303-961-3984

Email: dau9@nau.edu

Additional Editor Comments (if provided):

In Table S5, there are cases where no arthropods were reported for a given tree on a given sampling date. For example, there is no June 2018 record for tree J20 or J24. I assume these were the samples that were lost to flooding (line 135). I suggest inserting a row for each of these samples, indicating that they were lost to flooding and so there are no data to report.

We thank the editor for their suggestion and have corrected our table (now S6 Table). 

We note that Figure 1 in your submission contains map images which may be copyrighted. All PLOS content is published under the Creative Commons Attribution License (CC BY 4.0), which means that the manuscript, images, and Supporting Information files will be freely available online, and any third party is permitted to access, download, copy, distribute, and use these materials in any way, even commercially, with proper attribution. For these reasons, we cannot publish previously copyrighted maps or satellite images created using proprietary data, such as Google software (Google Maps, Street View, and Earth). For more information, see our copyright guidelines: http://journals.plos.org/plosone/s/licenses-and-copyright.

Our map image was created by us on PowerPoint and is not copyrighted. We note that we previously had images of highway signs and have replaced those in our figure with our own, which should now meet PLOS’s license requirements. 

Reviewer #1: 

In this manuscript, Uhey et al. examine the role of tree types, climate and productivity on the richness and abundance patterns of ground dwelling arthropods in Pinyon-Juniper woodlands. They sampled ground arthropods using pitfall traps at several sites underneath Pinyon and Juniper trees at several time points and use generalized linear mixed models and other methods to address their question. I think they posed a very interesting question. Often when we think about arthropod-plant interactions, we think of foliage-dwelling arthropods and not ground-dwelling arthropods. The authors did a really good job of explaining how ground dwelling arthropod communities could be affected by tree types in the introduction and they also collected a lot of arthropods. However, I was not entirely convinced by their sampling methods, at least as they are currently written. I think the results section needs more details as well. They used a whole host of statistical methods but do not report the results of these in sufficient detail in the manuscript. Finally, I think their discussion needs to be developed further. They left out some important papers that they should have cited and do not address the limitations of their current work or point to any future directions.

Major comments:

1. I am concerned about the sampling method used in the paper and whether the authors even had enough power to discern differences in the ground dwelling arthropod communities harbored by Pinyon vs Juniper trees. They only sampled under three Pinyon and three Juniper trees at each site, which I am not sure gives them enough power to be able to discern any differences between arthropod communities under those trees. They also chose isolated trees for sampling (also need to clarify how isolated these isolated trees were, i.e. how far from the nearest tree?). However, there might be confounding factors that lead to these trees being isolated that affect the arthropod community. I think a better study design would have been to record the density of juniper and pinyon trees in say 10 m2 radius around each pitfall trap and look at how variation in the density of juniper and pinyon trees in an area affects arthropod diversity. If they sampled an isolated juniper tree that was in an area dominated by Pinyon trees, then are they sampling ground-dwelling arthropods that might prefer juniper tree litter? I am not sure. I think they need to add more details about the relative abundance of Pinyon and Juniper trees around their pitfall traps. If all the sites had a similar density of Pinyon and Juniper trees around the pitfall traps, then I do not think they are able to test the effects of tree type at all.

We thank the reviewer for their concern about our sampling, we have addressed this with several additions and modifications in the manuscript which A) more thoroughly explains our sampling design and study system, B) adds data, explanation, and discussion of our tree composition both at the site and tree level, and C) adds evidence that tree/site combinations were adequately sampled. We elaborate below.

A) We have improved our description of our sampling design and study system on lines 120-138 in the methods and with photos in a new figure (Fig. 2). Most forests would be difficult to separate tree-species effects, as trees grow in close proximity. Lacking before was a description of why our woodlands are unique compared to forests, as wide-spacing and isolation of trees is common (Fig. 2). Our woodland system is therefore ideal in having large-areas influenced by either pinyon or juniper which both establish sprawling canopies and accumulate litter underneath. We have clarified that we purposefully choose large-canopied pinyons or junipers (>3m2 radius canopy, canopy diameters now listed under Supp. Table 1) that were not growing close (>5m2) of the opposing species to avoid confounding effects. Pit traps were placed near trunks of trees and we believe these samples to be good representatives of their respective tree species. 

B) To help contextualize our system, we have measured plots as suggested by the reviewer with estimations of tree composition at each site (tree density in five parallel belt transects of 100mX20m=100m2 plot) and at each tree (tree density in 10m2 radius plots) described in the methods on lines 134-138, added as Fig. 2., and data made available in S1 Table. The tree-level data reaffirms our sampling design (no opposing tree species within >5m2 radius, 5-10m2 radius with few/no opposing tree species). 

In regards to the reviewer’s concerns, we only isolated trees purposefully from opposing species; some trees were adjacent to the same species while others isolated. Roughly half of trees were isolated within a 5m2 radius and 16.7% isolated within a 10m2 radius (S1 Table). Tree composition did change across sites (Fig. 2). We feel our samples are representative of the variation found in PJ woodlands of our region and aren’t heavily influenced by confounding factors. However, we have added discussion on tree isolation and the limitations of our study on lines 424-426. 

C) While the sample size of three of each tree species at each site may appear low, we believe our sampling was intensive. Each tree was sampled for five separate week-long sampling periods (total of 15 samples per tree species at each site), with a large number of arthropods captured (as mentioned by the reviewer). To further show sampling was adequate, we have added species accumulation curves that show asymptotes in sampling for each tree/site combination (S7 Fig) referenced on lines 317-318. Moreover, statistical power is also a product of variation, which was low within compared to among sites (S7 Table). All models showed good fit statistics. 

2. I think it would have been interesting if the authors had also sampled foliage dwelling arthropods. That would have set up a cool comparison of how diversity patterns of foliage dwelling versus ground dwelling arthropods are affected by tree types, climate, and productivity. This might have also allayed some concerns about sampling adequacy if they had noticed an effect of tree types on foliage dwelling arthropods but not ground dwelling ones. Unless the authors already have these data, I know it might not be feasible to go and collect foliage dwelling arthropods now, but this is something they could include in the future directions section of the discussion.

We thank the reviewer for their insight which we hope to one-day test. While we did not sample foliage-dwelling arthropods, we agree it should be added to the future directions section of the discussion which now appears on lines 460-462.

3. In the first sentence of the study sites section in Methods, the authors say that they sampled along two elevational gradients spanning ~1000m. However, when I look at figure 1, it looks like all their sites are between 1900m-2200m elevation. If that is indeed the case, then the elevational gradient is just 300m. I think it is important for the authors to clarify this. I would also be careful when citing literature on elevational gradients and comparing their results to other studies on elevational gradients. This study only captures a subset of the entire gradient, so any patterns seen or not seen with regards to elevation need to be interpreted with that in mind. I am not suggesting that they should have sampled a longer gradient; given their research question with regards to Pinyon-Juniper woodlands, I think the elevational distribution of their study sites make a lot of sense. I am just suggesting a more nuanced explanation of results in terms of elevation.

We thank the reviewer, this was a typo and we have revised our methods to state our gradients span ~300m (or ~1,000ft) on line 93. We agree with adding nuance to our elevational comparisons and have contextualized our elevational findings in our discussion on lines 465-466. 

4. I am unclear about the specification of the GLMMs. Date and site were used as random effects, but were they entered as two separate random effects or were they treated as crossed random effects? As far as I understand their sampling design, they should have been treated as crossed random effects. Did the authors run model diagnostics to make sure that the model specification was okay? Also, why the use of AIC instead of AICc which corrects for small sample size? Lastly, I think they should include their GLMM table in the main paper instead of putting it in the supplement. At least the table for all arthropods. The model should include standard errors in addition to the slope values. That would help understand if they had enough statistical power. I would also encourage the authors to include their R script as a supplement.

We have taken the reviewer’s suggestions and improved our methodological description and reporting of our GLMM statistics. Date and site were crossed random effects, now stated in line 256 of the methods, along with a more detailed description of GLMM specification and model diagnostics (lines 256-258). We have included AICc values in S3 Table referenced on line 253 which agree with our previously reported AIC values. Our GLMM table now appears in the main text as Table 2 on lines 369-374, with standard errors reported. We’ve added our R script as supplement “S2 File”. 

5. I think the authors need to look through the literature more carefully and make sure to cite relevant studies. For example, I was wondering if there are other studies that show an effect of tree types on ground dwelling arthropods and a quick search led me to this paper: https://doi.org/10.1111/j.0030-1299.2008.15972.x I think this paper should be cited in this study. I would advise the authors to look through the literature once again and cite other relevant studies, it will make the discussion section much stronger. Also, like I said earlier, discussing the limitations of the study and what future work should be done in this field of research should be included in the discussion.

We thank the reviewer for their suggestion to improve the discussion and the provided resource. We have added a more thorough discussion on tree species effects on ground-dwelling arthropods on lines 403-463, based on additional literature (de Abreu Pestana et al. 2020, Černecká et al. 2020, Plowman et al. 2020, Pardon et al. 2019, Perry et al. 2018, Thorn et al. 2016, Lange et al. 2014, Pringle & Fox-Dobbs 2008, and Vehviläinen et al. 2008). We’ve also included more discussion of limitations and future work. 

de Abreu Pestana, L.F., de Souza, A.L.T., Tanaka, M.O., Labarque, F.M. and Soares, J.A.H., 2020. Interactive effects between vegetation structure and soil fertility on tropical ground-dwelling arthropod assemblages. Applied Soil Ecology, 155, p.103624.

Černecká, Ľ., Mihál, I., Gajdoš, P. and Jarčuška, B., 2020. The effect of canopy openness of European beech (Fagus sylvatica) forests on ground‐dwelling spider communities. Insect Conservation and Diversity, 13(3), pp.250-261.

Lange, M., Türke, M., Pašalić, E., Boch, S., Hessenmöller, D., Müller, J., Prati, D., Socher, S.A., Fischer, M., Weisser, W.W. and Gossner, M.M., 2014. Effects of forest management on ground-dwelling beetles (Coleoptera; Carabidae, Staphylinidae) in Central Europe are mainly mediated by changes in forest structure. Forest Ecology and Management, 329, pp.166-176.

Pardon, P., Reheul, D., Mertens, J., Reubens, B., De Frenne, P., De Smedt, P., Proesmans, W., Van Vooren, L. and Verheyen, K., 2019. Gradients in abundance and diversity of ground dwelling arthropods as a function of distance to tree rows in temperate arable agroforestry systems. Agriculture, ecosystems & environment, 270, pp.114-128.

Perry, K.I., Wallin, K.F., Wenzel, J.W. and Herms, D.A., 2018. Forest disturbance and arthropods: Small‐scale canopy gaps drive invertebrate community structure and composition. Ecosphere, 9(10), p.e02463.

Plowman, N.S., Mottl, O., Novotny, V., Idigel, C., Philip, F.J., Rimandai, M. and Klimes, P., 2020. Nest microhabitats and tree size mediate shifts in ant community structure across elevation in tropical rainforest canopies. Ecography, 43(3), pp.431-442.

Pringle, R.M. and Fox‐Dobbs, K., 2008. Coupling of canopy and understory food webs by ground‐dwelling predators. Ecology Letters, 11(12), pp.1328-1337.

Thorn, S., Bußler, H., Fritze, M.A., Goeder, P., Müller, J., Weiß, I. and Seibold, S., 2016. Canopy closure determines arthropod assemblages in microhabitats created by windstorms and salvage logging. Forest ecology and management, 381, pp.188-195.

Vehviläinen, H., Koricheva, J. and Ruohomäki, K., 2008. Effects of stand tree species composition and diversity on abundance of predatory arthropods. Oikos, 117(6), pp.935-943.

Minor comments:

I am personally not a huge fan of acronyms within papers unless they are already universally used. So, I think it is better to just say ground dwelling arthropods instead of GDAs and Pinyon-Juniper woodlands instead of PJs. I think it makes for easier reading, but it is up to you.

We appreciate the reviewer’s preference on acronyms and understand they can be cumbersome when not commonly used. However, ‘GDA’ (Meyer et al. 2015, Xiao et al. 2020, Hasin & Booncher 2020) and ‘PJ’ (Weppner et al. 2013, Carrol et al. 2016, Jahromi & Agblevor 2017, Hartsell et al. 2020, Filippelli et al. 2020) are both widely used and we believe convenient due to the repetition of these terms in our manuscript. We leave it to the editor’s preference though. 

Carroll, R.W., Huntington, J.L., Snyder, K.A., Niswonger, R.G., Morton, C. and Stringham, T.K., 2017. Evaluating mountain meadow groundwater response to Pinyon‐Juniper and temperature in a great basin watershed. Ecohydrology, 10(1), p.e1792.

Filippelli, S.K., Falkowski, M.J., Hudak, A.T., Fekety, P.A., Vogeler, J.C., Khalyani, A.H., Rau, B.M. and Strand, E.K., 2020. Monitoring pinyon-juniper cover and aboveground biomass across the Great Basin. Environmental Research Letters, 15(2), p.025004.

Hartsell, J.A., Copeland, S.M., Munson, S.M., Butterfield, B.J. and Bradford, J.B., 2020. Gaps and hotspots in the state of knowledge of pinyon-juniper communities. Forest Ecology and Management, 455, p.117628.

Hasin, S. and Booncher, K., 2020. Change in ground-dwelling arthropod communities in different agroecosystems in Wang Nam Khiao, Nakhon Ratchasima province, Thailand. Agriculture and Natural Resources, 54(2), pp.139-149.

Jahromi, H. and Agblevor, F.A., 2017. Upgrading of pinyon-juniper catalytic pyrolysis oil via hydrodeoxygenation. Energy, 141, pp.2186-2195.

Meyer, W.M., Eble, J.A., Franklin, K., McManus, R.B., Brantley, S.L., Henkel, J., Marek, P.E., Hall, W.E., Olson, C.A., McInroy, R. and Loaiza, E.M.B., 2015. Ground-dwelling arthropod communities of a sky island mountain range in southeastern Arizona, USA: Obtaining a baseline for assessing the effects of climate change. PloS one, 10(9).

Weppner, K.N., Pierce, J.L. and Betancourt, J.L., 2013. Holocene fire occurrence and alluvial responses at the leading edge of pinyon–juniper migration in the Northern Great Basin, USA. Quaternary Research, 80(2), pp.143-157.

Xiao, H., Du, C., Yuan, X. and Li, B., 2020. Multiple floods affect composition and community structure of the ground-dwelling arthropods in the riparian zone of the Three Gorges Reservoir. Ecological Indicators, 113, p.106220.

Line 39: need a comma between “tree specialization hypothesis, a specific version”

Agreed and revised as suggested by reviewer.

Line 40: “dominant” not “dominate”

Agreed and revised as suggested by reviewer.

Line 62: “management of these ecosystems”

Agreed and revised as suggested by reviewer.

Line 116: What was the source of this climate data? Did they have weather stations or is it satellite data from Worldclim? Please describe.

This was described at the end of the paragraph (data source is PRISM), but we understand that this may not read clearly, so have moved our statement on data source to follow our statement at the paragraph start (lines 159-160).

Line 124: Spatial resolution of 800m seems pretty low given the proximity of the sites that were sampled. I thought there was higher resolution data available for US, but I am not sure. MODIS seems to have data at 250m resolution(https://modis.gsfc.nasa.gov/data/dataprod/mod13.php) and has EVI in addition to NDVI which has been recommended in some recent work. I do not have expertise in this domain, but I would encourage the authors to look a little bit more into this.

While this spatial data is course it is best seen as an estimate for relative climate information for each site. Remke et al. 2020 showed relative temperature differences predicted by PRISM were similar to observed temperature differences by weather stations. Of course finer resolution data would be better, but MODIS and other imagining sources do not provide estimates of climate parameters averaged over long time periods. Numerous studies have used PRISM data as an estimate for climatic conditions (e.g. Delph et al. 2014, Youngsteadt et al. 2015, Welti et al. 2020). Strachan and Daly (2017) show that observed spatial variation in temperature is high and complex, however, currently spatially explicit data does not exists, so PRISM still provides a reasonable alternative until more stations are used to improve spatial interpolation models (http://www.prism.oregonstate.edu/documents/pubs/2017JGR_TestingPRISMTemperature_Strachan.pdf).

Delph, R.J., Clifford, M.J., Cobb, N.S., Ford, P.L. and Brantley, S.L., 2014. Pinyon pine mortality alters communities of ground-dwelling arthropods. Western North American Naturalist, 74(2), pp.162-184.

Remke, M.J., Hoang, T., Kolb, T., Gehring, C., Johnson, N.C. and Bowker, M.A., 2020. Familiar soil conditions help Pinus ponderosa seedlings cope with warming and drying climate. Restoration Ecology.

Strachan, S. and Daly, C., 2017. Testing the daily PRISM air temperature model on semiarid mountain slopes. Journal of Geophysical Research: Atmospheres, 122(11), pp.5697-5715.

Youngsteadt, E., Dale, A.G., Terando, A.J., Dunn, R.R. and Frank, S.D., 2015. Do cities simulate climate change? A comparison of herbivore response to urban and global warming. Global Change Biology, 21(1), pp.97-105.

Welti, E.A., Prather, R.M., Sanders, N.J., de Beurs, K.M. and Kaspari, M., 2020. Bottom‐up when it is not top‐down: Predators and plants control biomass of grassland arthropods. Journal of Animal Ecology.

Line 170: “Fig 3”, not “Fig 2”

Figure numbers have changed and are correct.

Line 177: There is a typo, it should be “Linear” not “Liner”

Agreed and revised as suggested by reviewer.

Line 204: Please clarify what you mean when you say ant data was relativized by maximum.

We have changed this analysis based on reviewer’s two comment and no longer modify ant data in this manner.

Figure 3: This figure might not be readable to people with red-green colorblindness. Please check and try to avoid the use of red and green together in a figure. This is a really useful resource to help pick colors that are colorblind friendly: https://colorbrewer2.org/#type=sequential&scheme=BuGn&n=3

Agreed and revised as suggested by reviewer using colorbrewer2.org.

Reviewer #2: 

In this study, the authors compare communities of ground dwelling arthropods in pinyon-juniper woodlants using specimens from pitfall traps that were repeatedly sampled over the course of two years. With an emphasis on ants and beetles, they ask whether there are distinct species assemblages based on tree type, and also evaluate the effects of climate and productivity as measured by NDVI. They find variable effects of precipitation, NDVI, and elevation depending on the taxon and metric. While I think the idea is interesting, I have three main concerns:

1. Mixed genera, morphospecies, and species.

The authors do not identify all taxa, instead frequently using morphospecies or genera. This is common and completely understandable. However, the grouping is rather uneven, such that the analyses are performed on communities composed of a rather haphazard mixture of species, morphospecies, and genera. This makes me rather skeptical of the results. For example, several speciose genera (e.g., Crematogaster) are left as genera ('Crematogaster sp.'), while other smaller genera are identified to species (e.g., Forelius). I recognize that these genera are very difficult to identify, but I do not think it is valid to treat 'Crematogaster sp' as taxonomically equivalent to Forelius mccooki in terms of community composition and dissimilarity. This applies to other arthropods as well. For example, 'Melanotus similis' is treated as a separate taxon compared to 'Melanotus sp', but most specimens are identified only to genus. I expect that the taxa identified to species may have an outsized weight on the analyses, despite representing a minority of the actual species, while genera like Crematogaster sp. or Formica sp. may be obscuring differences between the communities since many species are treated as one. Obviously it would be best to identify all specimens to species (or to morphospecies representing a few likely species for those that may not be separable). Otherwise, identifying only to genera might be a better approach.

We appreciate the reviewer’s concern regarding level of taxonomic identity and have addressed this with several additions and revisions to our manuscript. First, during the review and revision process additional identifications of eight morphospecies were received from experts which we have updated in S6 Table (Crematogaster punctulata, Formica (neogatates complex), Myrmecocystus mimicus, Myrmica rugiventris, Crocidema arizonicus, Stachyocenmus apicalis, and Listrus senilis). Second, we have added to S6 Table rows detailing who assigned species/morphospecies identifications and links to photographs of voucher specimens posted on bugguide.net where possible. Third, we have added to our results clearer statements on numbers of species/morphospecies designations on line 315. Including the new identifications, 81% of specimens are identified to species. 15% of specimens are left at morphospecies, but 16 of those morphospecies are singleton specimens and thereby must represent single species. This level of identification is high compared to most studies investigating ground-dwelling arthropod communities (Delph et al. 2014, Meyer et al. 2015, Yekwayo et al. 2018, Ferrenberg et al. 2019, Ferreira et al. 2020).

We feel confident that morphospecies designations are reasonably representative of single species, as experts determined voucher specimens of each species/morphospecies per site. Of course it is always possible for a morphospecies designation to represent multiple cryptic species; as the reviewer points out this is a common problem among arthropod studies. Morphospecies was assigned for certain groups that are difficult to identify or in need of taxonomic revision by experts who confirmed multiple voucher specimens of morphospecies were most likely the same species. 

We found few differences with ground-dwelling arthropod communities between tree species when analyzed on the species/morphospecies level, doing our analysis on genera rather than species/morphospecies (i.e. losing taxonomic information) could only make communities look more similar. We believe our taxonomic depth high in comparison to other studies and this data supports robust analyses, however we have given this issue more recognition in our manuscript by adding discussion points on morphospecies and diversity limitations on lines 414-416. 

Delph, R.J., Clifford, M.J., Cobb, N.S., Ford, P.L. and Brantley, S.L., 2014. Pinyon pine mortality alters communities of ground-dwelling arthropods. Western North American Naturalist, 74(2), pp.162-184.

Ferreira, P.M., Andrade, B.O., Podgaiski, L.R., Dias, A.C., Pillar, V.D., Overbeck, G.E., Mendonça Jr, M.D.S. and Boldrini, I.I., 2020. Long-term ecological research in southern Brazil grasslands: Effects of grazing exclusion and deferred grazing on plant and arthropod communities. PloS one, 15(1), p.e0227706.

Ferrenberg, S., Wickey, P. and Coop, J.D., 2019. Ground-Dwelling Arthropod Community Responses to Recent and Repeated Wildfires in Conifer Forests of Northern New Mexico, USA. Forests, 10(8), p.667.

Meyer, W.M., Eble, J.A., Franklin, K., McManus, R.B., Brantley, S.L., Henkel, J., Marek, P.E., Hall, W.E., Olson, C.A., McInroy, R. and Loaiza, E.M.B., 2015. Ground-dwelling arthropod communities of a sky island mountain range in southeastern Arizona, USA: Obtaining a baseline for assessing the effects of climate change. PloS one, 10(9).

Yekwayo, I., Pryke, J.S., Gaigher, R. and Samways, M.J., 2018. Only multi-taxon studies show the full range of arthropod responses to fire. PloS one, 13(4), p.e0195414.

2. Treatment of ant worker abundances

The authors acknowledge that abundances of ant workers in their samples were biased by proximity of some pitfalls to ant nests, and they accounted for this by relativizing worker abundances by the maximum. However, this does not adequately or correctly account for the relevant peculiarities of ant life histories. In particular, ant foraging behavior varies by species, and some species are more likely than others to form foraging trails (Lanan 2014) that might lead to large numbers of ants from the same colony being captured in one trap (Bestelmeyer et al 2000). With ants in pitfall traps, it is unfortunately not possible to use the observed worker abundances as reliable estimates of either local worker abundance or local colony density. There are incidence-based metrics available that could be useful. A simple alternative would be to use a presence-based metric like Jaccard dissimilarity rather than Bray-Curtis (analyses in lines 206–207).

We thank the reviewer for improving our analysis; we have taken the suggestion to redo our ordination based-analyses with an incidence-based Jaccard dissimilarity metric. While this did not significantly change any result, we are glad to report more conservative estimates of ant communities which do vary considerably as the reviewer states. We have updated our methods on lines 295-297 and all figures/tables impacted by the change. 

3. Discussion topics

The discussion should also be expanded. For example, it would be good to discuss the spatial scales involved in the context of tree specificity. Ground-dwelling arthropods generally move quite a bit, so the assumption you're making is that abundance in pitfall traps is correlated with 'time spent in that microhabitat'. Isolated trees were also intentionally selected for sampling to avoid mixed canopies of pinyon and juniper. Would this affect the observed community? What if the taxa that are specialized to tree type are also less likely to travel across larger stretches of open canopy? Would you expect differences between ants and beetles, given that ants are central place foragers with a (generally) stationary nest, unlike beetles?

We thank the reviewer for these excellent suggestions for discussion points. We have extensively rewritten and added much to our discussion, including the reviewer’s suggested topics. We have also identified arthropods to functional group, which has improved contextualizing our results in the discussion.

As just an additional comment, this dataset seems to be very well-suited to using an occupancy model to account for species that were present but happened to not be sampled, with interesting combinations across aspect, elevation, and tree type. I recognize this is not the focus of this study, but it could be an interesting avenue for the authors to explore in the future.

This is an interesting idea we hope to pursue in the future.

References:

Bestelmeyer, B. T., Agosti, D., Alonso, L. E., Brandão, R. F., Brown, W. L. J., Delabie, J. H. C., & Silvestre, R. (2000). Field Techniques for the Study of Ground-Dwelling Ants: An Overview, Description, and Evaluation. In D. Agosti, J. D. Majer, L. E. Alonso, & T. R. Schultz (Eds.), Ants: Standard Methods For Measuring and Monitoring Biodiversity (pp. 122–144). http://antbase.org/ants/publications/20339/20339.pdf

Lanan, M. (2014). Spatiotemporal resource distribution and foraging strategies of ants (Hymenoptera: Formicidae). Myrmecological News, 20, 53–70.

Minor items

27–28: briefly summarise these points here

We agree and have summarized our discussion points in the lines 27-30 of the abstract. 

40: "dominant"

Agreed and revised as suggested by reviewer.

46–47: Doesn't the preceding sentence contradict this claim? If there are few studies which are largely limited to birds, how can you state that the shifting composition is changing animal communities?

As written these statements were contradictory, we have revised our second sentence to a speculation (“may”) from statement (“is”). That is, now our introduction states that the shifting composition of pinyon-juniper woodlands may change animal communities. The uncertainty is caused by limited studies comparing pinyon and juniper animal communities, setting up our study.

48: maybe something like "Arthropods often closely associate with vegetation types"

Agreed and revised as suggested by reviewer.

49: specify 'plant' or 'tree' genotypes

Agreed and revised as suggested by reviewer.

56: Add "The" to the start of the sentence to make it a bit more clear

Agreed and revised as suggested by reviewer.

60–62: This sentence is rather vague. Could you tie it to the previous sentence more, specify how preservation and management could be improved, and/or add an example with a reference?

We have replaced our sentence on lines 69-71 to be more specific and tie into the previous sentence with an added a reference (Schowalter 2017). 

Schowalter, T., 2017. Arthropod diversity and functional importance in old-growth forests of North America. Forests, 8(4), p.97.

89: 'bitterbrush'

Agreed and revised as suggested by reviewer.

197–198: RStudio is just the IDE / interface - I think only the R version is necessary here.

Agreed, we have removed the RStudio reference. 

221: Specify which Supp Table

Agreed and revised to specify S5 Table. 

332: I think here (and throughout) it would be good to qualify 'diversity' and 'richness', as readers will generally assume these are at the species level.

Clarified on lines 316-317. 

Fig. 4: I think it would be clearer to show the dates as different colors and the trees as different shapes

Agreed and changed as suggested by reviewer.

---

## [Decision Letter · Decision Letter 1]

21 Jul 2020

PONE-D-20-11956R1

Ground-dwelling Arthropods of Pinyon-Juniper Woodlands: Arthropod Community Patterns are Driven by Climate and Overall Plant Productivity, not Host Tree Species

PLOS ONE

Dear Dr. Uhey,

Thank you for submitting your manuscript to PLOS ONE. After careful consideration, we feel that it has merit but does not fully meet PLOS ONE’s publication criteria as it currently stands. Therefore, we invite you to submit a revised version of the manuscript that addresses the points raised during the review process.

The reviewers and I agree that this revised version of the manuscript is much improved from your initial submission. Reviewers 1 and 2 listed a handful of remaining minor concerns that you should address. I also have my own list of editorial or grammatical edits for you to consider (see "Additional Editor Comments", below). If you address all of these comments, then the manuscript should be suitable for publication.

I received another comment from someone other than the three reviewers of record on your submission. To summarize, this individual argued that you should have used a repeated measures ANOVA approach to analyze your data, rather than generalized linear mixed models. I disagree. Repeated measures approaches, at least in this context, tend to violate the sphericity assumption, and GLMMs offer better flexibility. Nevertheless, it may be worth including a brief sentence that explains why you chose to use GLMMs for your tests of richness and abundance. I'll leave that up to you.

We look forward to receiving your revised manuscript.

Kind regards,

Frank H. Koch, PhD

Academic Editor

PLOS ONE

Additional Editor Comments (if provided):

Line 38 - suggest inserting a comma after "woodlands"

Line 44 - suggest inserting a comma after "properties (9,10)"

Line 46 - suggest inserting a comma after "pinyons"

Lines 94-95 - the San Francisco Peaks are a mountain range, i.e., there is no singular San Francisco Peak. The listed elevation is for Humphreys Peak, which is the tallest peak in the range.

Line 100 - capitalize "peaks"

Line 102 - "The relative proportion of pinyons and junipers..." - there are two relative proportions, one for pinyons and another for junipers.

Line 169 - replace "collectivity" with "collectively", or maybe "together"

Line 311 - replace "habit" with "habitat"

Line 328 - replace "are" with "may", or delete "be"

Line 354 - insert commas after "gradient" and after "woodlands"

Line 359 - replace "affect" with "affects"

Line 361 - replace "can't" with "cannot"

Line 373 - insert hyphen between "Low" and "elevation"

Line 375 - make "Ponderosa" lower-case

Line 401 - replace "highlight" with "highlighted"

Reviewers' comments:

Reviewer's Responses to Questions

**Comments to the Author**

1. If the authors have adequately addressed your comments raised in a previous round of review and you feel that this manuscript is now acceptable for publication, you may indicate that here to bypass the “Comments to the Author” section, enter your conflict of interest statement in the “Confidential to Editor” section, and submit your "Accept" recommendation.

Reviewer #1: (No Response)

Reviewer #2: (No Response)

Reviewer #3: All comments have been addressed

2. Is the manuscript technically sound, and do the data support the conclusions?

Reviewer #1: Yes

Reviewer #2: Partly

Reviewer #3: Yes

3. Has the statistical analysis been performed appropriately and rigorously? 

Reviewer #1: Yes

Reviewer #2: No

Reviewer #3: Yes

4. Have the authors made all data underlying the findings in their manuscript fully available?

Reviewer #1: No

Reviewer #2: Yes

Reviewer #3: Yes

5. Is the manuscript presented in an intelligible fashion and written in standard English?

Reviewer #1: Yes

Reviewer #2: Yes

Reviewer #3: Yes

6. Review Comments to the Author

Reviewer #1: I commend the authors on doing a really thorough job with the revisions. I just noticed a few minor errors/typos that I am listing here. My only other concern is the data availability. The authors say the data is available to download as supplements or from Symbiota Collection of Arthropod Network, but no link is provided leaving the reader to locate the data themselves, which can be cumbersome. The authors included the R output on my request, but without the csv file, it is difficult to replicate their analyses. So, I would encourage the authors to make the data available in a more easily accessible format. And here are the other minor edits: 1. Line 263, I think it should say "a significant indicator" instead of "significant indicators"; 2. Line 296: Table 2 does not show that NDVI is a significant predictor of ant richness, so I think the authors should remove NDVI from this sentence; 3. Line 311, "habitat" instead of "habit"; 4. Line 328: these is an extra "be"; 5. Line 330, "exacerbated" instead of "exasperated"

Reviewer #2: The authors have greatly improved the manuscript in this draft, and they have adequately responded to the majority of my previous comments. I do still have a few concerns, but I think they should be easy to address.

First, I am still not satisfied with the handling of ant abundances. I realize it seems like I might be harping on this, but it really is a difficult quantity to measure. The authors run GLMMs for abundance of 1) all GDA, 2) only ants, and 3) only beetles. In Table 2, the estimates for all GDAs and for ants are quite similar, which makes sense because 76% of the individuals were ants. While ants are certainly a very abundant and important component of the GDA community, the observed abundances are not representative given the issues with pitfall traps, particularly with regards to samples along gradients. For example, Camponotus vicinus is not 250x as abundant under tree 135 compared to tree 166, but that is the data going into the model.

Ant workers collected in pitfall traps simply do not represent either colony abundance or worker abundance, and using worker counts as estimates of abundance will (frankly and unfortunately) result in nonsense. Using the worker abundances from pitfall traps directly just does not give the kind of information needed to estimate abundances. My suggestion would be to instead limit the abundance GLMMs to non-ant GDAs, and show results for 'Beetle abundance' and 'Other GDA abundance', where 'other' is non-ant non-beetle taxa. This is only an issue for abundances - richness should be unaffected.

Second, unless time or computing power are limiting factors, it is generally preferable to fit all possible models and compare using AICc rather than using backwards stepwise selection (208-209). However, I think there must be something missing or I am misunderstanding the analyses, as I cannot find any GLMM results for subsets of the predictor variables representing the final models. If it is the former, the results should be included and I would contend that Table 2 should only show the final models. If it is the latter, this section and the results perhaps need some clarifying.

A final minor suggestion for improved reproducibility the future: R notebooks (in RStudio) are a really nice way to provide the code and output. PDFs can be generated easily, with text, code, and output inline without the additional characters that come from pasting into a .docx file (S2). In my opinion, it makes it easier for readers, for anyone interested in trying/adapting your code, and even for the authors.

# Line items

41: 'genus'

46-49: I still feel like this needs more qualification or justification... Maybe first rephrase to state the differences found in the bird studies, or even that they found a difference, and then qualify the second sentence with something like 'if other taxa show similar microhabitat specialization, ...'

200-211: Is it correct that 'abundance' for ants here refers to the number of workers? For clarity, it would be worth stating below (230-232) that incidence was used *only* in the similarity matrices (e.g., "In contrast to the above analyses which directly used the number of ant workers collected...").

Reviewer #3: (No Response)

7. PLOS authors have the option to publish the peer review history of their article (what does this mean?). If published, this will include your full peer review and any attached files.

Reviewer #1: **Yes: **K. Supriya

Reviewer #2: No

Reviewer #3: **Yes: **Heather Slinn

---

## [Author Response · Author response to Decision Letter 1]

5 Aug 2020

Dear Editor,

 Below are our responses (in red) to the reviewers comments for the manuscript, “Ground-dwelling Arthropods of Pinyon-Juniper Woodlands: Arthropod Community Patterns are Driven by Climate and Overall Plant Productivity, not Host Tree Species”, catalog number PONE-D-20-11956. 

We appreciate the feedback provided from both reviewers and the editor which has strengthened our manuscript. We have accepted all suggestions and revised our manuscript accordingly. All authors have approved the changes.

Thank you for your time and consideration,

Derek Uhey, Corresponding Author

Northern Arizona University

College of Engineering, Forestry and Natural Sciences

200 E. Pine Knoll Dr.

Flagstaff, AZ 86011 USA

Ph: 303-961-3984

Email: dau9@nau.edu

Editor Comments:

The reviewers and I agree that this revised version of the manuscript is much improved from your initial submission. Reviewers 1 and 2 listed a handful of remaining minor concerns that you should address. I also have my own list of editorial or grammatical edits for you to consider (see "Additional Editor Comments", below). If you address all of these comments, then the manuscript should be suitable for publication.

I received another comment from someone other than the three reviewers of record on your submission. To summarize, this individual argued that you should have used a repeated measures ANOVA approach to analyze your data, rather than generalized linear mixed models. I disagree. Repeated measures approaches, at least in this context, tend to violate the sphericity assumption, and GLMMs offer better flexibility. Nevertheless, it may be worth including a brief sentence that explains why you chose to use GLMMs for your tests of richness and abundance. I'll leave that up to you.

We thank the editor for their suggestion and understanding of our choice of statistical method. We have added a brief statement on why we choose the GLMM approach on lines 209-210 of the methods. 

Additional Editor Comments (if provided):

Line 38 - suggest inserting a comma after "woodlands"

Line 44 - suggest inserting a comma after "properties (9,10)"

Line 46 - suggest inserting a comma after "pinyons"

Lines 94-95 - the San Francisco Peaks are a mountain range, i.e., there is no singular San Francisco Peak. The listed elevation is for Humphreys Peak, which is the tallest peak in the range.

Line 100 - capitalize "peaks"

Line 102 - "The relative proportion of pinyons and junipers..." - there are two relative proportions, one for pinyons and another for junipers.

Line 169 - replace "collectivity" with "collectively", or maybe "together"

Line 311 - replace "habit" with "habitat"

Line 328 - replace "are" with "may", or delete "be"

Line 354 - insert commas after "gradient" and after "woodlands"

Line 359 - replace "affect" with "affects"

Line 361 - replace "can't" with "cannot"

Line 373 - insert hyphen between "Low" and "elevation"

Line 375 - make "Ponderosa" lower-case

Line 401 - replace "highlight" with "highlighted"

We have accepted all of the above suggestions and thank the editor for these improvements.

Reviewer #1: I commend the authors on doing a really thorough job with the revisions. I just noticed a few minor errors/typos that I am listing here. My only other concern is the data availability. The authors say the data is available to download as supplements or from Symbiota Collection of Arthropod Network, but no link is provided leaving the reader to locate the data themselves, which can be cumbersome. The authors included the R output on my request, but without the csv file, it is difficult to replicate their analyses. So, I would encourage the authors to make the data available in a more easily accessible format. 

We thank the reviewer and agree data availability can be improved. We have included all relevant CSV files as supplements (in zip file with R-script) and improved our R-script (S2 File). Unfortunately, our records in Symbiota Collection of Arthropod Network are only searchable by specimen label information. The search platform is found at the given link. We also include links to photographed specimens at bugguide.net with url #s in S6 Table. With the changes to our S2 file, along with our many supplemental materials, we believe our data is easily available. 

And here are the other minor edits: 

1. Line 263, I think it should say "a significant indicator" instead of "significant indicators"; 

2. Line 296: Table 2 does not show that NDVI is a significant predictor of ant richness, so I think the authors should remove NDVI from this sentence; 

3. Line 311, "habitat" instead of "habit"; 

4. Line 328: these is an extra "be"; 

5. Line 330, "exacerbated" instead of "exasperated"

We have revised our manuscript and accepted the reviewer’s suggested line comments. 

Reviewer #2: The authors have greatly improved the manuscript in this draft, and they have adequately responded to the majority of my previous comments. I do still have a few concerns, but I think they should be easy to address.

First, I am still not satisfied with the handling of ant abundances. I realize it seems like I might be harping on this, but it really is a difficult quantity to measure. The authors run GLMMs for abundance of 1) all GDA, 2) only ants, and 3) only beetles. In Table 2, the estimates for all GDAs and for ants are quite similar, which makes sense because 76% of the individuals were ants. While ants are certainly a very abundant and important component of the GDA community, the observed abundances are not representative given the issues with pitfall traps, particularly with regards to samples along gradients. For example, Camponotus vicinus is not 250x as abundant under tree 135 compared to tree 166, but that is the data going into the model.

Ant workers collected in pitfall traps simply do not represent either colony abundance or worker abundance, and using worker counts as estimates of abundance will (frankly and unfortunately) result in nonsense. Using the worker abundances from pitfall traps directly just does not give the kind of information needed to estimate abundances. My suggestion would be to instead limit the abundance GLMMs to non-ant GDAs, and show results for 'Beetle abundance' and 'Other GDA abundance', where 'other' is non-ant non-beetle taxa. This is only an issue for abundances - richness should be unaffected.

We thank the reviewer for their concerns. The accuracy of estimating ant-abundances via pit-traps is certainly debatable, given we found no significant patterns with ant or GDA abundance, we agree our non-results should be accompanied with more discussion of potential methodological bias. We believe it is still important to report our non-results on the abundance of ants and over-all GDAs but have added discussion on the short-comings of pit-traps for estimating abundance on lines 415-420. Further, we have added the ‘other’ group to our analysis as suggested by the reviewer. 

Second, unless time or computing power are limiting factors, it is generally preferable to fit all possible models and compare using AICc rather than using backwards stepwise selection (208-209). However, I think there must be something missing or I am misunderstanding the analyses, as I cannot find any GLMM results for subsets of the predictor variables representing the final models. If it is the former, the results should be included and I would contend that Table 2 should only show the final models. If it is the latter, this section and the results perhaps need some clarifying.

We thank the reviewer for their suggestion and have updated our methods and results to strengthen our GLMM results. We fit all possible models for each GDA response variable and include AICc comparisons in S3 Table. Our methods are updated to reflect this on lines 207-221. We have taken the reviewer suggestion to have table 2 only include final models.

A final minor suggestion for improved reproducibility the future: R notebooks (in RStudio) are a really nice way to provide the code and output. PDFs can be generated easily, with text, code, and output inline without the additional characters that come from pasting into a .docx file (S2). In my opinion, it makes it easier for readers, for anyone interested in trying/adapting your code, and even for the authors.

We are grateful for this suggestion and have included an R-notebook output in our S2 files along with CSV datasheets.

# Line items

41: 'genus'

Revised.

46-49: I still feel like this needs more qualification or justification... Maybe first rephrase to state the differences found in the bird studies, or even that they found a difference, and then qualify the second sentence with something like 'if other taxa show similar microhabitat specialization, ...'

We have revised these sentences as suggested by reviewer. 

200-211: Is it correct that 'abundance' for ants here refers to the number of workers? For clarity, it would be worth stating below (230-232) that incidence was used *only* in the similarity matrices (e.g., "In contrast to the above analyses which directly used the number of ant workers collected...").

We have clarified that use of incidence-based ant abundance was only in similarity matrices for ordination analysis. 

Reviewer #3: (No Response)

---

## [Editor Report · Decision Letter 2]

13 Aug 2020

Ground-dwelling Arthropods of Pinyon-Juniper Woodlands: Arthropod Community Patterns are Driven by Climate and Overall Plant Productivity, not Host Tree Species

PONE-D-20-11956R2

Dear Dr. Uhey,

We’re pleased to inform you that your manuscript has been judged scientifically suitable for publication and will be formally accepted for publication once it meets all outstanding technical requirements.

Kind regards,

Frank H. Koch, PhD

Academic Editor

PLOS ONE

Additional Editor Comments (optional):

Thank you for responding to all of the comments on the previous revision. Your manuscript is now suitable for publication, although I have some minor editorial comments, listed below:

Line 40: insert “a” before “host tree species”

Line 45: insert comma after “pinyons”

Line 48: “taxa groups” seems odd – maybe “animal taxa”?

Line 60: replace “describe” with “have described”

Line 109: “ponderosa” should be lower-case

Line 115: “chose” instead of “choose”

Lines 140-141: Did you mean to say “30-year averages of annual average precipitation…”? For precipitation, I think you’re probably working with a 30-year average of the annual total. On the other hand, it is reasonable to have a 30-year average of the annual average temperature.

Line 146: Was the spatial resolution 250 m? 250 m2 would mean each raster cell was about 16 m on a side.

Line 149: Here, you denote the raster package using single quotes. In line 221, you denote the mvabund packages using italics. In lines 216 and 229, you don’t distinguish the names of the lme4, arm, labdsv, and indicspecies packages, and the same is true in line 246 for the vegan and ecodist packages. Pick one approach and use it consistently.

Lines 192-193: Vapor pressure is represented using yellow in Fig 3; precipitation is represented using blue.

Line 199: “had” instead of “has”

Line 208: Insert “the” before “Akaike” (since it’s a singular criterion).

Line 244: “analyses” instead of “analysis”

Line 273: “differed significantly”

Line 284: “increased significantly”

Line 310: insert “the” before “Bray-Curtis”

Line 318: “forest types” instead of “forests”

Line 381: delete “arthropods” (redundant)

Line 384: “GDA” instead of “GDAs”

S1 Site descriptions.xlsx, “100X20m plots” tab – “parallel” is misspelled
---

## [Editor Report · Acceptance letter]

17 Aug 2020

PONE-D-20-11956R2 

Ground-dwelling Arthropods of Pinyon-Juniper Woodlands: Arthropod Community Patterns are Driven by Climate and Overall Plant Productivity, not Host Tree Species 

Dear Dr. Uhey:

I'm pleased to inform you that your manuscript has been deemed suitable for publication in PLOS ONE. Congratulations! Your manuscript is now with our production department. 

Kind regards, 

on behalf of

Dr. Frank H. Koch 

Academic Editor

PLOS ONE